# A Fractal Model of Hydraulic Conductivity for Saturated Frozen Soil



**Lei Chen [1,2] , Dongqing Li [1], Feng Ming [1,]*, Xiangyang Shi [1,2] and Xin Chen [1,2]**

1   State Key Laboratory of Frozen Soil Engineering, Northwest Institute of Eco-Environment and Resources, Chinese Academy of Sciences, Lanzhou 730000, China; ChenLei8@lzb.ac.cn (L.C.); dqli@lzb.ac.cn (D.L.); shixiangyang2008@163.com (X.S.); chenxcumt@sina.com (X.C.)
2   University of Chinese Academy of Sciences, Beijing 100049, China
*   Correspondence: mingfeng05@lzb.ac.cn; Tel.: +86-93-1496-7469

**Abstract:** In cold regions, hydraulic conductivity is a critical parameter for determining the water flow in frozen soil. Previous studies have shown that hydraulic conductivity hinges on the pore structure, which is often depicted as the pore size and porosity. However, these two parameters do not sufficiently represent the pore structure. To enhance the characterization ability of the pore structure, this study introduced fractal theory to investigate the influence of pore structure on hydraulic conductivity. In this study, the pores were conceptualized as a bundle of tortuous capillaries with different radii and the cumulative pore size distribution of the capillaries was considered to satisfy the fractal law. Using the Hagen-Poiseuille equation, a fractal capillary bundle model of hydraulic conductivity for saturated frozen soil was developed. The model validity was evaluated using experimental data and by comparison with previous models. The results showed that the model performed well for frozen soil. The model showed that hydraulic conductivity was related to the maximum pore size, pore size dimension, porosity and tortuosity. Of all these parameters, pore size played a key role in affecting hydraulic conductivity. The pore size dimension was found to decrease linearly with temperature, the maximum pore size decreased with temperature and the tortuosity increased with temperature. The model could be used to predict the hydraulic conductivity of frozen soil, revealing the mechanism of change in hydraulic conductivity with temperature. In addition, the pore size distribution was approximately estimated using the soil freezing curve, making this method could be an alternative to the mercury intrusion test, which has difficult maneuverability and high costs. Darcy's law is valid in saturated frozen silt, clayed silt and clay, but may not be valid in saturated frozen sand and unsaturated frozen soil.

**Keywords:** frozen soil; soil freezing curve; hydraulic conductivity; fractal model; Darcy's law

## 1. Introduction

Hydrology in cold regions is a rapidly progressing research field, representing an important sub-discipline of hydrology. Recently, research interest in cold region hydrology has been spurred by global warming-induced hydrological and ecological changes to cold regions, such as permafrost degradation, glacial recession and surface runoff shrinkage [1]. Cold region hydrology includes permafrost hydrology, glacier hydrology and snow hydrology, of which permafrost hydrology is the broader scientific field owing to permafrost's wide distribution, covering about a quarter of the land surface of the world [2]. Frozen soil is a special type of soil-water system composed of soil particles, liquid water, ice and gas. The emergence and existence of ice changes the driving force of water migration, leading to changes in migration direction, migration distance, migration velocity and migration amount; these produce further changes in the water distribution in frozen soil. Previously,

frozen soil has often been conceptualized as an impermeable barrier that inhibits infiltration and promotes surface and near-surface runoff; however, researchers have since investigated the hydraulic conductivity of frozen soil and found that frozen soil is not a complete aquiclude, as liquid water could migrate in the frozen soil. Thus, water migration in frozen soil is one of the main processes in permafrost hydrology. Due to the free energy of soil particles and water-air interfaces, some water in frozen soil remains unfrozen at sub-zero temperatures [3]—defined as unfrozen water. The unfrozen water will flow along a temperature gradient or pressure gradient, resulting in water and solute redistribution [4], frost heaving [5], and salt expansion [6]. Such damage causes problems during the construction of engineering ground structures in cold regions, including roads, railways, pipelines, buildings and dams. In frozen soil, at least in high-temperature frozen soil, the unfrozen water transported in the frozen soil is always assumed to follow Darcy's law:

$$V = k_\mathrm{f} \frac{\Delta H}{L} \tag{1}$$

where $V$ is Darcy's flux (m/s); $k_\mathrm{f}$ is the hydraulic conductivity of frozen soil (m/s); $\Delta H$ is the water head loss between two cross sections (m), $\Delta H = H_1 - H_2$, where $H$ is the total head (m), defined as the sum of the elevation head $Z$ and the pressure head $h_w$; and $L$ is the distance of the seepage path between these two cross sections (m). As Equation (1) shows, hydraulic conductivity is one of the key parameters for determining water flow in frozen soil.

To further predict water redistribution, frost heave and salt expansion, several hydrology and hydrogeology models had been developed by coupling a groundwater flow equation with heat transfer equations. Researchers applying these mathematical models are faced with difficulties due to a lack of precise values of hydraulic conductivity. The main reason for this deficiency is that measuring the hydraulic conductivity of frozen soil remains difficult and a complete expression for hydraulic conductivity is not yet available. This problem leads to a general assumption being made that the hydraulic conductivity of frozen soil is, in most cases, zero. However, it has been found in most of the studies that water flow still occurs in frozen soil, especially in high-temperature frozen soil, making this assumption unreasonable. The main objective of this paper is to provide a solution to this problem by presenting a new mathematical model for estimating the hydraulic conductivity of saturated frozen soil with a limited set of data.

## 2. Background

Many attempts have been undertaken to determine the hydraulic conductivity of frozen soil. Burt and Williams [7] measured saturated hydraulic conductivity using a permeameter. In this study, water was used as the fluid. To prevent the water from melting the frozen soil, lactose was added to ensure that the fluid was in thermodynamic equilibrium with the water in the soil. In addition, the lactose had restricted entry to the soil through the use of a dialysis membrane. In other studies, to solve the problem of the water melting the frozen soil, different fluids other than water have been used to measure hydraulic conductivity; for example, Horiguchi and Miller [8] used distilled water; Wiggert et al. [9] used dacane, octane, heptane and NaCl brine; McCauley et al. [10] used diesel; and Seyfried and Murdock [11] used air. All of these studies tended to overestimate hydraulic conductivity because the fluids could flow due to air-filled porosity, making it difficult to obtain an accurate hydraulic conductivity measuring by such experiments alone.

Many mathematical models for predicting the hydraulic conductivity of frozen soil have since been recommended. These mathematical models include empirical models and statistical models. For empirical models, there are three main types. The first type is determined by the assumption that the hydraulic conductivity of frozen soil is a function of temperature. For the convenient application in numerical models, Horiguchi and Miller [12] gave an empirical formula, $k_\mathrm{f} = C \times T^D$, where, $k_\mathrm{f}$ is the hydraulic conductivity of frozen soil (m/s), $T$ is the temperature (°C), and $C$ and $D$ are constant fitting parameters. Nixon [13] proposed a similar temperature-dependent empirical formula

as follows, $k_f = k_0/(-T)^\delta$, where, $k_0$ is the hydraulic conductivity at $-1\,°C$ and $\delta$ is the slope of the $k_f - T$ relationship in a log-log coordinate; however, these models must be determined using some type of experimental data, thus limiting their applicability. The second type of empirical model is determined by the assumption that the ice in saturated frozen soil and the air in unsaturated unfrozen soil plays a similar role in the process of water seepage when there is the same water content; that is the hydraulic conductivity of saturated frozen soil is closed to that of unsaturated melted soil. A drawback of this model is that it always overrates the hydraulic conductivity of frozen soil. The third model type is determined by the fact that the second model overestimates the hydraulic conductivity of frozen soil, resulting in an ice impedance being introduced. Jame and Norum [14] introduced an impedance factor, $k_f = k_u/10^{-E\theta_i}$, where, $k_u$ is the hydraulic conductivity of unfrozen soil with the same liquid water content (m/s), $E$ is an empirical constant and, $\theta_i$ is the volumetric content of ice (m$^3$/m$^3$). Based on agreement with experimental data from Jame and Norum [14] and Burt and Williams [7], Taylor and Luthin [15] gave a specific impedance factor, $k_f = k_u/10^{10\theta_i}$. Mao et al. [16] presented another impedance factor form to express hydraulic conductivity, $k_f = k_u \times (1 - \theta_i)^3$; however, many researchers have criticized this model because of the arbitrary choice of impedance factor. Shang et al. [17] suggested that the impedance factor would be better determined using $I = k_f/k_u$; however, this method is still limited by a paucity of the measured data and, moreover, the impedance factor is not a constant.

Meanwhile, several statistical models have also been proposed, of which there are two main types. The first type of statistical model is determined by the capillary bundle model and the Hagen-Poiseuille equation. Watanabe and Flury [18] developed a new capillary bundle model, in which the ice is assumed to form in the centre of the capillaries, leaving a circular annulus open for liquid water flow. On this premise, a newly developed hydraulic conductivity model was suggested, as seen in Equation (2):

$$k_f = \frac{\gamma_w \pi}{8\mu\tau} \sum_{J=k+1}^{M} n_J \left[ R_J{}^4 - r_{i_J}{}^4 + \frac{\left(R_J{}^2 - r_{i_J}{}^2\right)^2}{\ln\left(r_{i_J}/R_J\right)} \right] \quad R_J - d(T) \geq r_i(T)$$
$$k_f = \frac{\gamma_w \pi}{8\mu\tau} \sum_{J=k+1}^{M} n_J R_J{}^4 \quad R_J - d(T) < r_i(T) \tag{2}$$

where, $n_J$ is the number of capillaries of radius $R_J$ per unit area; $r_{i_J}$ is the radius of cylindrical ice (m); $d(T)$ is the thickness of the water film (m); $\mu$ is the dynamic viscosity of the fluid (kg/m·s); $\tau$ is tortuosity; $M$ is the number of different capillary size classes and $k = 0, 1, 2, \cdots$, is an index for each decrease of water content. In this model, the soil water characteristic curve is used to determine the hydraulic conductivity of frozen soil, but the soil water characteristic curve is the constitutive equation in the unfrozen soil. Similarly, Weigert and Schmidt [19] proposed another model for predicting unsaturated hydraulic conductivity of partly frozen soil. The model considered that the effective saturated hydraulic conductivity of partly frozen soil equals the difference between that of saturated melted soil and unsaturated melted soil. Thus, the hydraulic conductivity of saturated melted soil equals the sum of the hydraulic conductivity of each class of capillaries, which is proportional to $r_x{}^2\Delta\theta_x$; therefore, the model was determined, as shown in Equation (3):

$$k_f = k_s - k_u$$
$$k_u = \sum_{J=k+1}^{M} k_s \frac{r_x{}^2\Delta\theta_x}{r_1{}^2\Delta\theta_1 + r_2{}^2\Delta\theta_2 + r_3{}^2\Delta\theta_3 + \cdots + r_J{}^2\Delta\theta_J} \tag{3}$$

where, $k_u$ and $k_s$ represent unsaturated and saturated hydraulic conductivity, respectively (m/s); $r_x$ is the mean radius (m); and $\Delta\theta_x$ is the share of the pore class $x$ of the total volumetric water content (cm$^3$/cm$^3$). The second type of statistical model is determined by the hydraulic conductivity model for melted soil and the Clausius-Clapeyron equation for describing the phase change process between ice and water. The hydraulic conductivity of frozen soil is converted by the hydraulic conductivity of unsaturated melted soil and the Clausius-Clapeyron equation. Azmatch [20] and

Tarnawski [21] present their models to predict the hydraulic conductivity of frozen soil based on different hydraulic conductivity models of melted soils, although this method tends to overestimate hydraulic conductivity [3].

Given the problems with the above methods, there is great need for a new approach to estimating hydraulic conductivity. Since fractal theory (see Section 3) was first created, this method has been applied to different areas to determine the permeability of porous material; for example, Yu [22,23] proposed a fractal permeability model for fabrics and bi-dispersed porous media, predicting the corresponding permeability well. Yao [24] investigated the fractal characteristics of coals using the mercury porosimetry method, and analysed the influence of the fractal dimension of pores on the permeability of coals. Pia [25] presented an intermingled fractal units model based on fractal theory for the purpose of predicting the permeability of porous rocks. Xu [26] used fractal geometry to describe soil pores and obtained the fractal dimension of pores from porosimetric measurements. A permeability function was eventually put forward to predict the permeability of soils. The fractal theory used by this study to estimate the permeability performed well for the chosen materials. The pore size distribution of frozen soil was confirmed as satisfying the fractal characteristics [27–29] required; therefore, we attempted to apply the fractal theory to determine its hydraulic conductivity.

In the present study, a fractal model for hydraulic conductivity of saturated frozen soil was presented. A method for determining the model parameters was also proposed. Model performance was subsequently evaluated by comparing predictions with experimental data from the existing literature and other models. Furthermore, the validity of Darcy's law in saturated frozen soil was explained.

## 3. Fractal Model for Hydraulic Conductivity of Frozen Soil

### 3.1. Fractal Theroy

In classical Euclidean geometry, the dimensions of ordered objects are the integers; for example, the dimensions of points, straight lines, planes, and volumes are 0, 1, 2 and 3, respectively. However, for disordered objects such as irregular lines, planes or volumes, Euclidean geometry is not applicable. To solve this problem, fractal geometry has been introduced. In fractal geometry, the dimension of fractal objects is not an integer, but a fraction. Normally, $1 < D < 2$ in two-dimensional space and $2 < D < 3$ in three-dimensional space. Fractal theory emerges based on fractal geometry. Fractal theory describes and studies the objects from the perspective of fractal dimensions and its mathematical methods. The fractal theory broke out of the traditional barriers of one-dimensional lines, two-dimensional surfaces, three-dimensional spaces, bringing it closer to a description of the real properties and states of complex systems, more consistent with the diversity and complexity of objects.

### 3.2. The Fractal Capillary Bundle Model of Frozen Soil

The real size or geometric shape of pores are difficult to describe, meaning pores are sometimes visualized as circular capillary tubes [30]. A capillary bundle model has consequently been proposed and frequently applied [31–33]. Although the model is different from real soils, it contains many of the same properties and so can be used to analyse the water flow in frozen soil [18]. As illustrated in Figure 1, the pores are conceptualised as an assembly of tortuous capillary tubes.

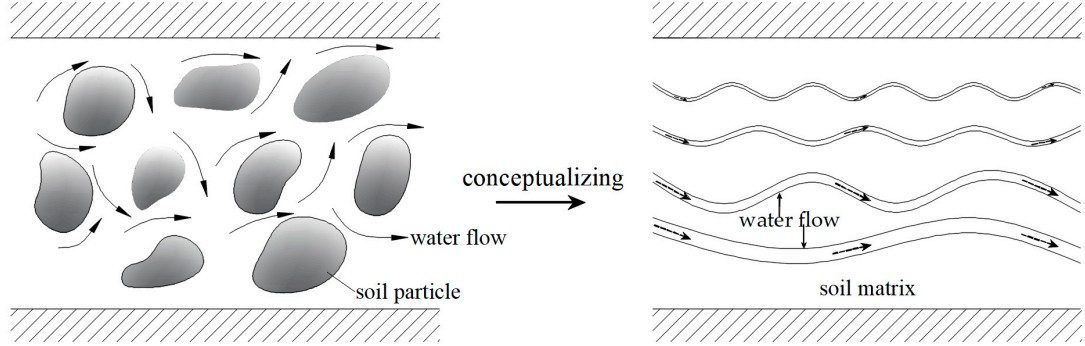

**Figure 1.** Schematic of capillary bundle model of frozen soil.

To develop the fractal capillary bundle model, we assumed the frozen soil was homogeneous and isotropic; the solid matrix was incompressible; the pores could be treated as capillary tubes with different radii; the capillaries were initially filled with liquid water and the water flow in the different capillary tubes remain independent; The freezing of pore water started from macrospores and the freezing temperature of pore water reduced with the decrease of pore diameter. Figure 2 gives a schematic of the freezing process in capillary tubes, where the fully saturated tubes are filled with liquid water in the unfrozen state. As the temperature decreases, the liquid water in larger tubes are frozen into ice and a thin liquid film remains. The radii of ice filled tubes and the thickness of the liquid film both depend on temperature.

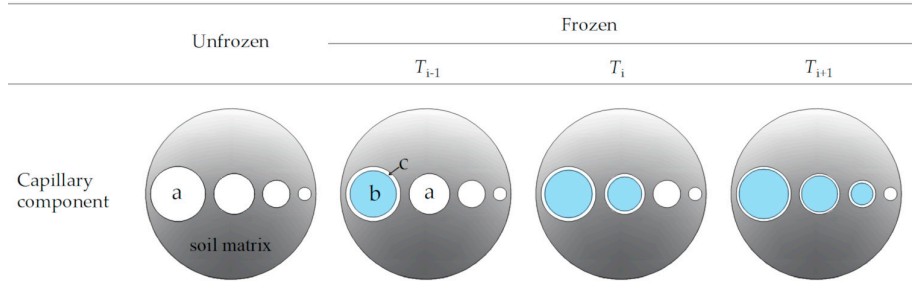

<sup>a</sup>Water filled capillary tubes; <sup>b</sup>Ice filled capillary tubes; <sup>c</sup>Film of water

**Figure 2.** Schematic of freezing process of capillary tubes.

*3.3. Derivation of the Fractal Model to Determine the Hydraulic Conductivity of Frozen Soil*

The cumulative pore size distribution follows the fractal scaling law [34]; therefore the cumulative number of pores satisfied:

$$N(\geq r) = \left(\frac{r_{\max}}{r}\right)^D \tag{4}$$

where $N$ is the cumulative number of pores with a radius greater than or equal to $r$; $r_{\max}$ is the maximum radius (m) and, $D$ is the pore size dimension, $1 < D < 2$. Differentiating Equation (4) with respect to $r$ yields:

$$-dN(\geq r) = D r_{\max}{}^D r^{-(D+1)} dr \tag{5}$$

Equation (5) gives the pore number between the pore size $r$ and $r + dr$. In Equation (5), $-dN > 0$, which indicates that pore number decreases with an increase in pore size.

The soil pore system is assumed to be a bundle of capillary tubes with different radii; therefore, the Hagen-Poiseuille equation can be used to describe the water flow in the soil, as given by [22,35]:

$$q = r^4 \frac{\pi}{8} \frac{\rho_w g}{\eta} \frac{\Delta H}{L_e} \tag{6}$$

where $q$ is the water flow rate in a single capillary ($m^3/s$); $r$ and $L_e$ are the radius and length of the capillary, respectively (m); $\eta$ is the dynamic viscosity of the fluid (Pa·s); $\Delta H$ is the hydraulic gradient; $\rho_w$ is the density of water ($kg/m^3$); and $g$ is gravitational acceleration ($m/s^2$). The total flow rate $Q$ ($m^3/s$) can be obtained by integrating the single flow rate, $q$, from the minimum pore radius to the maximum radius with the aid of Equation (6):

$$Q = -\int_{r_{min}}^{r_{max}} q dN = \frac{\pi \rho_w g}{8\eta} \frac{\Delta H}{L_e} \frac{D}{4-D} r_{max}^4 \left[ 1 - \left( \frac{r_{min}}{r_{max}} \right)^{4-D} \right] \tag{7}$$

Since $1 < D < 2$, the exponent $4 - D > 2$, $r_{min}/r_{max} < 10^{-2}$; therefore, $(r_{min}/r_{max})^{4-D} \ll 1$. Then Equation (7) is reduced to:

$$Q = \frac{\pi \rho_w g}{8\eta} \frac{\Delta H}{L_e} \frac{D}{4-D} r_{max}^4 \tag{8}$$

The average velocity $J_a$ (m/s) through the capillaries can be obtained by dividing $Q$ by the total pore area $A_p$ ($m^2$):

$$J_a = \frac{\pi \rho_w g}{8\eta A_p} \frac{\Delta H}{L_e} \frac{D}{4-D} r_{max}^4 \tag{9}$$

The total pore area in Equation (9) can be expressed as [36]:

$$A_p = \int_{r_{min}}^{r_{max}} \pi r^2 (-dN) = \frac{\pi D (1-\varepsilon) r_{max}^2}{2-D} \tag{10}$$

where $\varepsilon$ is the porosity. Inserting Equation (10) into Equation (9) yields:

$$J_a = \frac{\rho_w g}{8\eta} \frac{\Delta H}{L_e} \frac{2-D}{4-D} \frac{r_{max}^2}{1-\varepsilon} \tag{11}$$

The Darcy flux $J_T$ can be determined by using the Dupuit-Forchheimer equation:

$$J_T = \varepsilon J_a = \frac{\rho_w g}{8\eta} \frac{\Delta H}{L_e} \frac{\varepsilon}{1-\varepsilon} \frac{2-D}{4-D} r_{max}^2 \tag{12}$$

where $\tau$ is a tortuosity, which is the ratio between the length of soil column $L$ (m) and the length of the capillary $L_e$ (m). Comparing Equation (12) with Darcy's law, $J_T = k\frac{\Delta H}{L}$, results in the expression for the hydraulic conductivity as follows:

$$k = \frac{\rho_w g}{8\eta \tau} \frac{\varepsilon}{1-\varepsilon} \frac{2-D}{4-D} r_{max}^2 \tag{13}$$

where $k$ is the hydraulic conductivity. For a straight capillary bundle fractal model ($\tau = 1$), the hydraulic conductivity is reduced to:

$$k = \frac{\rho_w g}{8\eta} \frac{\varepsilon}{1-\varepsilon} \frac{2-D}{4-D} r_{max}^2 \tag{14}$$

It can be found from Equation (13) that for water flow through a given soil sample, the hydraulic conductivity is related to the maximum pore size, pore size dimension, and tortuosity. Of these parameters, maximum pore size plays a key role in affecting hydraulic conductivity. These parameters reflect the pore structure characteristic of the soil, and each of them has a clear physical meaning, which reveals a significant advantage of the model.

In the model, ice is assumed to form first in the largest water-filled tubes upon freezing based on the Gibbs-Thomson effect. The Gibbs-Thomson effect describes how a depression in the freezing point is inversely proportional to pore size. Due to the free energy of the capillary wall, some water in the

vicinity of the capillary wall remains unfrozen at subzero temperatures; hence why the frozen capillary is considered to be composed of an ice column and a thin liquid film in the vicinity of the tube wall. As the solid phase is assumed to be incompressible, the radius of the capillary cannot be affected by the frost heave of the water in the capillary. Therefore, the radius of the capillary is measured as the sum of the radius of the ice column and the thickness of the thin liquid film. The radius of the ice column can be assessed by the Gibbs-Thomson equation [37]:

$$r_i{}^i = \frac{2\sigma_{sl}}{L_f \cdot \rho_i} \frac{T_m}{T_m - T^i} \tag{15}$$

where, $r_i{}^i$ (m) is the critical radius of ice column at $T^i$ (°C); $\sigma_{sl}$ is the ice-liquid water interfacial free energy, $\gamma = 0.0818$ J/m²; $L_f$ is latent heat of fusion, $L = 334.56$ kJ/kg; $\rho_i$ is the density of ice, $\rho_i = 917$ kg/m³; and $T_m$ is the freezing temperature of bulk water given in Kelvin, $T_m = 273.15$ K. The thickness of the liquid water film can be assessed by [18]:

$$d^i = \left[ -\frac{A_H}{6\pi\rho_i L} \frac{T_m}{T_m - T^i} \right]^{1/3} \tag{16}$$

where $d^i$ is the film thickness of unfrozen water (m) and, $A_H$ is the Hamaker constant, $A_H = -10^{-19.5}$ J. The soil freezing curve (SFC) describes how the volumetric content of unfrozen water decreases with a decrease in the sub-zero temperature. The number of capillaries of radius $r^i$ can be determined using the SFC. As shown in Figure 3, the SFC is divided into several different spaced water-content intervals of width $\Delta\theta_u{}^i$, according to the experimental points, and the temperature $T\left(\theta_u{}^i\right)$ associated with the decrease in $\theta_u{}^i$ can be determined. All capillaries of radius $r \geq r^i$ are assumed to be frozen when $T = T^i$. Thus, the number of capillaries $N^i$ frozen at $T^i$, is equivalent to the increase in ice content in real soil, $\Delta V_i{}^i / \pi r_i{}^i L_e$, where $\Delta V_i{}^i$ is the increment of ice volume at $T^i$. $\Delta V_i{}^i$ can be calculated from the SFC, then the cumulative number of capillaries $N(r \geq r^i)$ is given by summing the number of capillaries $N^i$ of radius $r^i$ when $r \geq r^i$.

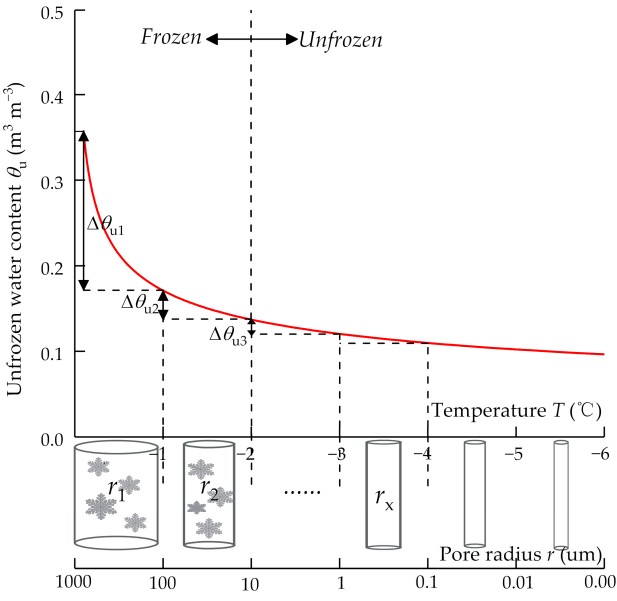

**Figure 3.** Scheme of a physical approach to determine the pore size distribution of frozen soil.

The pore size dimension was determined by the accumulative pore size distribution. The cumulative pore size distribution of fractal capillaries follows the fractal power law; therefore, taking the logarithm of both sides of Equation (4) gives:

$$D = \frac{\lg N}{\lg(r_{\max}/r)} \tag{17}$$

As seen in Equation (17), the pore size dimension was the slope of the accumulative pore size distribution in the double logarithmic coordinates, which was approximately a straight line. For frozen soil, the accumulative pore size distribution changed with temperature, leading to pore size dimension changes with temperature. The pore size dimension of frozen soil at different subzero temperatures needed to be determined using the accumulative pore size distribution at the corresponding temperature, the calculation of which had already been described above. In Equation (13), the dynamic viscosity of the water can be given as [18]

$$\eta = \eta_0 \exp(\frac{c}{T}) \tag{18}$$

where $\eta_0 = 9.62 \times 10^{-7}$ Pa · s and $c = 2046$ K. Equation (18) yields $\mu = 10^{-3}$ Pa · s when $T = 21.5\,°C$. The tortuosity can be assessed using the following relationship [34,35,38]

$$\tau = 1 + 0.41 \ln(1/\varepsilon) \tag{19}$$

In this model, the porosity is the ratio between the total unfrozen volume of capillaries and the volume of the soil column. As the unfrozen capillaries are assumed to be completely filled with unfrozen water, so the volume of unfrozen water can be conceived as the volume of the capillaries; thus, the porosity is equivalent to the volumetric content of unfrozen water. To illustrate the fractal hydraulic conductivity modeling process well, a flow diagram is given in Figure 4.

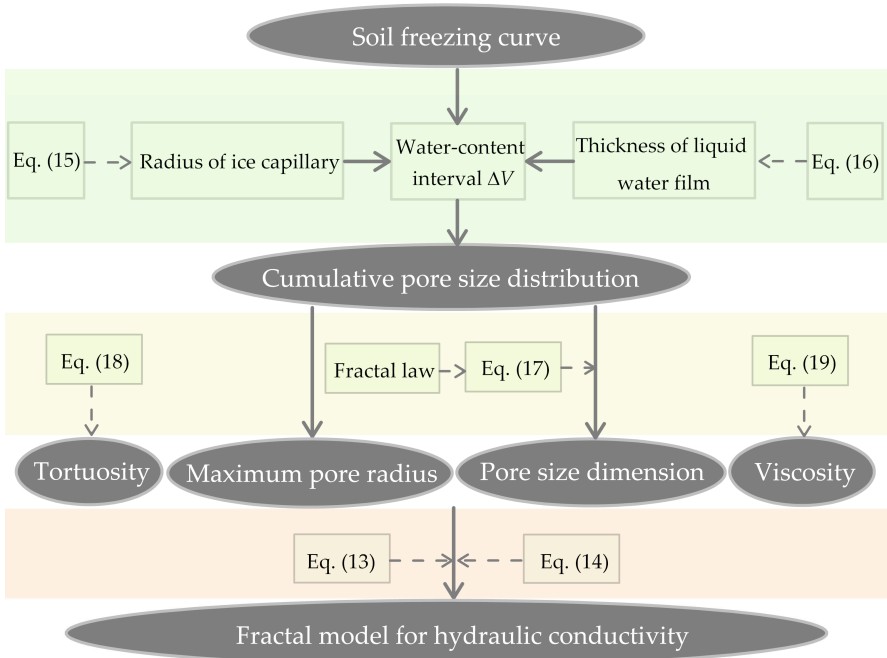

**Figure 4.** Flow diagram of fractal model building.

## 4. Methods

The present model was applied to existing literature on the measured hydraulic conductivity of saturated frozen soil to test its validity. Some existing studies did measure saturated conductivity; however, only a few simultaneously report the SFC, initial volumetric content and the dimension of the soil column, which are required by the model. Thus, model testing was limited by the paucity of the literature.

Burt and Williams [7] and Tokoro et al. [39] measured saturated hydraulic conductivity using self-developed experimental apparatus; their apparatus allows water to flow as a fluid. A summary of the physical parameters of the soil samples is given in Table 1.

**Table 1.** Physical parameters of the soil samples.

| Soil Type | Length (cm) | Diameter (cm) | Dry Density (g/m$^3$) | Initial Water Content (m$^3$ m$^{-3}$) |
|---|---|---|---|---|
| Silt [7] | 3 | 3.8 | 1.52 | 0.50 |
| clayey silt [7] | 3 | 3.8 | 3.8 | 0.33 |
| Silt [39] | 3 | 7 | 7 | 0.38 |

Figure 5 presents the SFCs of the three soil samples. As shown, the freezing curve decreases dramatically near 0 °C, after which the freezing curve became gradual. This result was because the proportion of capillaries with small radii increased as temperature decreased.

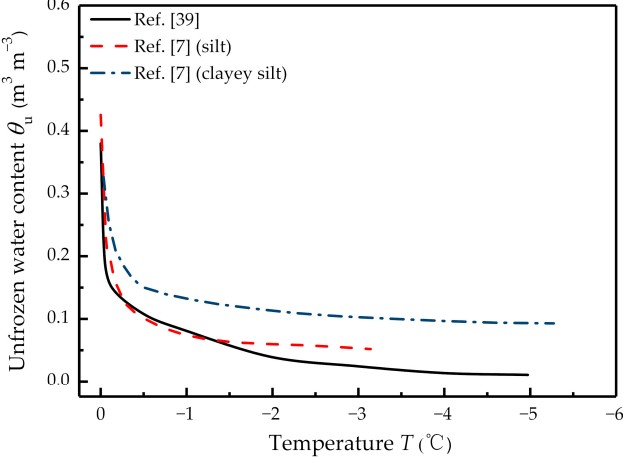

**Figure 5.** Soil freezing curve.

## 5. Results

### 5.1. Cumulative Pore Size Distribution Curve

Here, the experimental data from Tokoro et al. [39] were used as an example to describe the calculation process for finding the hydraulic conductivity of frozen soil. Figure 6 presents the cumulative pore size distribution estimated by the SFC, with the solid line representing the cumulative volume of pores and the dashed line representing the cumulative number of pores. The number of pores with small radii was markedly higher than the number with large radii. As the temperature decreased, the maximum radius decreased which led to changes in the pore size distribution curve. Additionally, the cumulative pore size distribution (the dashed line plotted in the double logarithmic coordinates) is approximately a straight line, the slope of which was the pore size dimension.

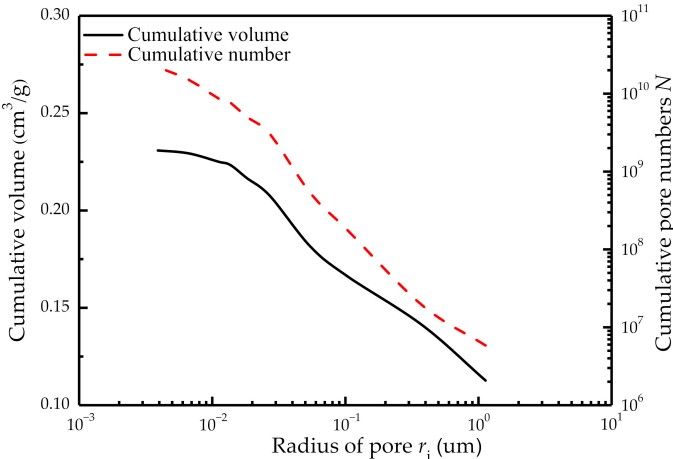

**Figure 6.** Pore size distribution determined by the soil freezing curve.

## 5.2. Maximum Pore Size, the Pore Size Dimension and Tortuosity

Figure 7a shows the relationship between the maximum pore radius and temperature, which was calculated by Equations (15) and (16). The maximum pore radius decreased with temperature. Figure 7b gives the relationship between the tortuosity and temperature, obtained using Equation (19). The tortuosity increased with temperature. Figure 7c presents the relationship between the pore size dimension and temperature, which was computed from the cumulative pore size distribution using Equation (17). The pore size dimension decreased linearly with temperature. As shown in the figure, the three parameters all varied with temperature, which caused the hydraulic conductivity to be affected by the temperature according to Equation (13).

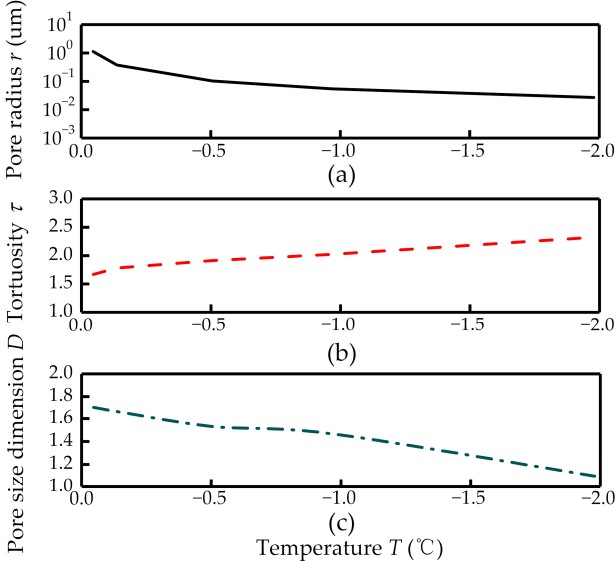

**Figure 7.** Variation of pore size dimension (**a**), maximum pore radius (**b**) and tortuosity (**c**)′with temperature.

## 5.3. Fractal Values of Hydraulic Conductivity

Figure 8a presents the hydraulic conductivity predicted by the present model for different pore size dimensions from Equations (13) and (14). Figure 8b shows the hydraulic conductivity predicted by the present model for different maximum pore radii from Equations (13) and (14). The hydraulic conductivity changed dramatically with the decrease in pore size dimension and maximum pore radius. This phenomenon can be explained as: the decrease of pore size and pore size dimension

causing a decrease in the number of capillaries, as well as the increase in the tortuosity causing an increase in water flow paths, leading to a decrease in flow rate. Of all these influencing factors, pore size plays a role in affecting hydraulic conductivity.

In the same way, the hydraulic conductivities of the three soil samples were predicted by the present model for different temperature and volumetric water content from Equation (13). The results are shown in Figure 8c,d. From Figure 8c, the hydraulic conductivity changed dramatically within $-1\ °C$, with similar results observed by Burt et al. [7] and Miller et al. [8]. These results can be explained by the fact that the decrease in temperature caused a decrease of the pore size and pore size dimension and increase of the tortuosity as shown in Figure 7, leading to a decrease in the hydraulic conductivity. Figure 8d shows the relationship between the hydraulic conductivity and volumetric water content, calculated from Equation (13). Figure 8d shows that the hydraulic conductivity increases with an increased in unfrozen water content.

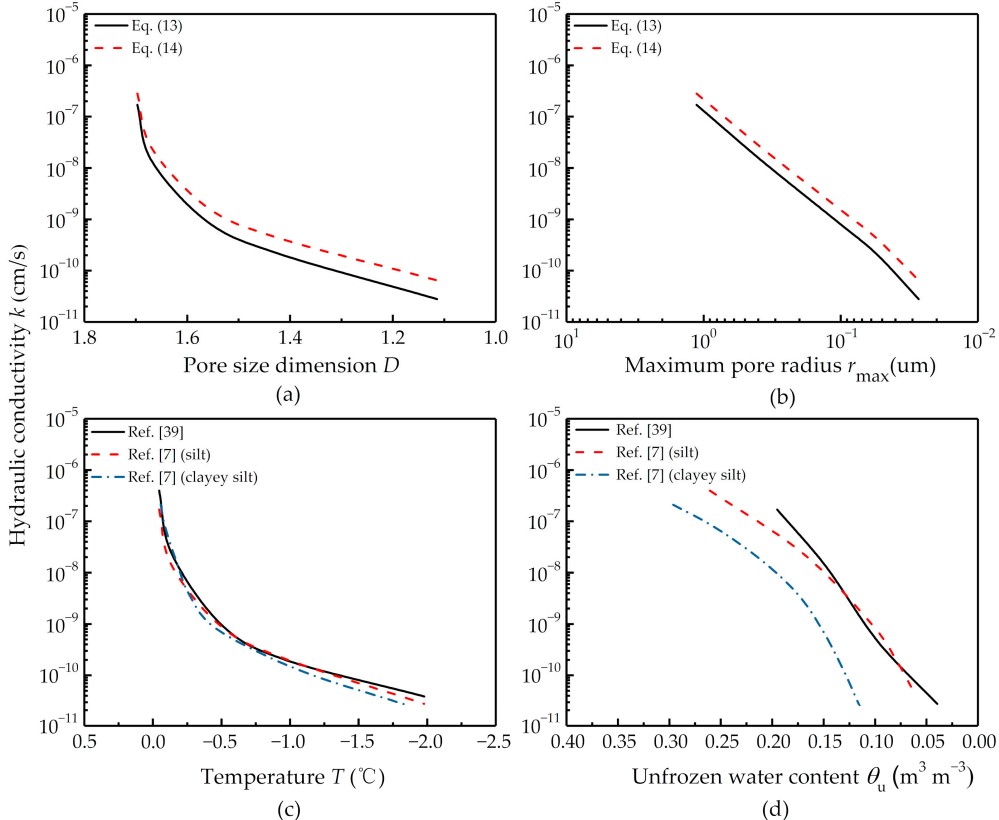

**Figure 8.** The effect of pore size dimension (**a**), maximum pore radius (**b**), temperature (**c**) and unfrozen water content (**d**) on hydraulic conductivity.

## 6. Discussion

### 6.1. Model Application

Figure 9 presents the comparison between the fractal model and the experimental data, with the symbols representing the experimental data, and the dashed line and dashed-dot line representing the predictions by the present models calculated from Equations (13) and (14), respectively. In both cases, the present model was in close agreement with the experimental data. From Figure 9a, using Equation (13), the present model shows good agreement with the experimental data except for the data point at 0.25 °C; however, using Equation (14), the present model overrated the experimental data. Figure 9b shows that using Equation (14), the present model showed good agreement with the experimental data while using Equation (13), the present model underestimate the experimental data. Figure 9c indicates the present model showed a slight difference from the experimental data; however,

the behavior of the experimental data is approximately captured by the present model. Overall, the present model based on fractal theory showed close agreement with the experimental data. Thus, the validity of the present model was verified.

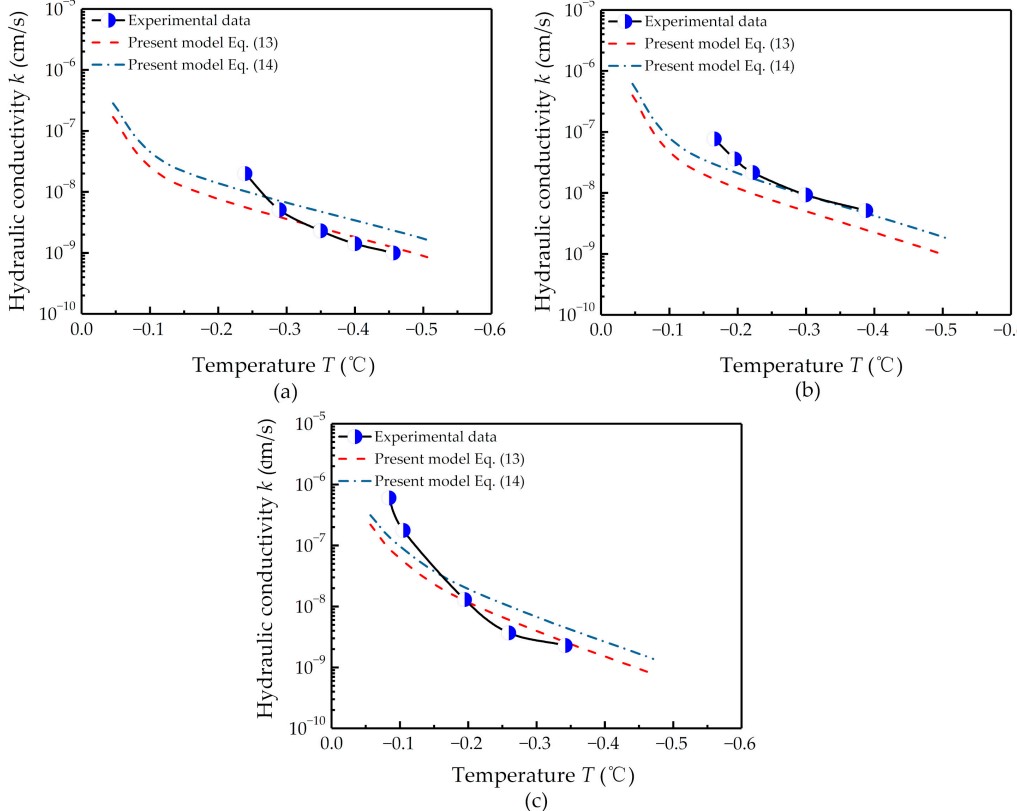

**Figure 9.** Comparison of the model results and experimental data. (**a**) Tokoro et al. [39] (silt); (**b**) Burt and Williams [7] (silt); (**c**) Burt and Williams [7] (clayey silt).

The differences between the predictions and experimental data can be translated into quantifiable terms by means of the mean squared error, MSE, which is an indicator of the overall magnitude of the residuals. The MSE is computed as follows

$$\text{MSE} \;=\; \frac{1}{N}\sum_{i=1}^{N}\left[\ln(k_{e,i}) - \ln(k_{p,i})\right]^{2} \tag{20}$$

where $N$ is the number of experimental data points; $k_{e,i}$ is the $i$th measured hydraulic conductivity (m/s) and $k_{p,i}$ is the $i$th predicted hydraulic conductivity (m/s). Table 2 summarises the MSE for the three data sets. As shown, the present model–using Equations (13) and (14)—made the best prediction for the Tokoro's [39] silt data set and Burt and Williams' [7] silt data set, respectively.

**Table 2.** Model prediction statistics for various data sets.

| Reference | Present Model | Mean Squared Error | Squared Error | | | | |
|---|---|---|---|---|---|---|---|
| | | | $-0.25\,°\text{C}$ | $-0.3\,°\text{C}$ | $-0.35\,°\text{C}$ | $-0.4\,°\text{C}$ | $-0.45\,°\text{C}$ |
| Tokoro et al. [39] | Equation (13) | 0.34 | 1.62 | 0.08 | 0 | 0.001 | 0.01 |
| | Equation (14) | 0.41 | 0.55 | 0.08 | 0.33 | 0.54 | 0.55 |
| Burt and Williams (silt) [7] | Equation (13) | 0.91 | 2.23 | 1.02 | 0.55 | 0.30 | 0.47 |
| | Equation (14) | 0.21 | 0.84 | 0.21 | 0.02 | 0 | 0 |
| Burt and Williams (clayey silt) [7] | Equation (13) | 0.98 | 3.50 | 1.19 | 0 | 0.22 | 0 |
| | Equation (14) | 0.78 | 1.98 | 0.41 | 0.16 | 0.97 | 0.40 |

As an additional test, the present fractal model was further compared with past models. The following three models were considered: the Nixon model [13], Mao et al. model [16] and Jame and Norum model [14]. Here, the experimental data from the study by Tokoro [39] were used as an example. In the Nixon model, $k_0$ and $\delta$ were obtained using the $k_f - T$ relationship in a log-log coordinate. The value of $k_0$ and $\delta$ were $2.0 \times 10^{-10}$ cm/s and $-2.273$, respectively. In the Mao et al. model and the Jame and Norum model, the value of $k_u$ was $8.1 \times 10^{-3} \times \theta_u^{6.21}$ cm/s and $\theta_i$ values were calculated from the SFC, $\theta_i = \theta_w \rho_w / \rho_i$. In the Jame and Norum model, the impedance factor was determined by the empirical constant $E$. To avoid an arbitrary choice of impedance factor, the empirical constant $E$ was determined using the method proposed by Shang et al. [17]. The experimental data at $-0.25\,°\text{C}$, $-0.3\,°\text{C}$ and $-0.35\,°\text{C}$ were used to determine the empirical constants, respectively. The corresponding empirical constants $E$ were $-0.48$, $-2.82$ and $-3.66$, respectively.

Figure 10 presents a visual comparison of various models for the silt data set from Tokoro [39]. The inferences on model performance can be drawn from model predictions for the experimental data. As shown, the Nixon model predicted the experimental data reasonably. This is ascribed to the fact that the model was determined from experimental data. Generally, the Mao et al. model overrated the experimental data. For the Jame and Norum model, when $E = -0.48$; the model deviated greatly from the experimental data; however, a significant improvement was observed when $E = -2.82$ and $E = -3.66$. It was found that the Jame and Norum model changed dramatically with the impedance factor. This shortcoming leads to a difficulty in accurately predicting the experimental data. Using Equation (13) in the present model showed much closer agreement with the experimental data. Moreover, the present model using Equation (13) is close to the Nixon model. Using Equation (13) in the present model with Equation (13) overrated the experimental data overall; this may be attributed to the fact that the water flow path in the soil was tortuous.

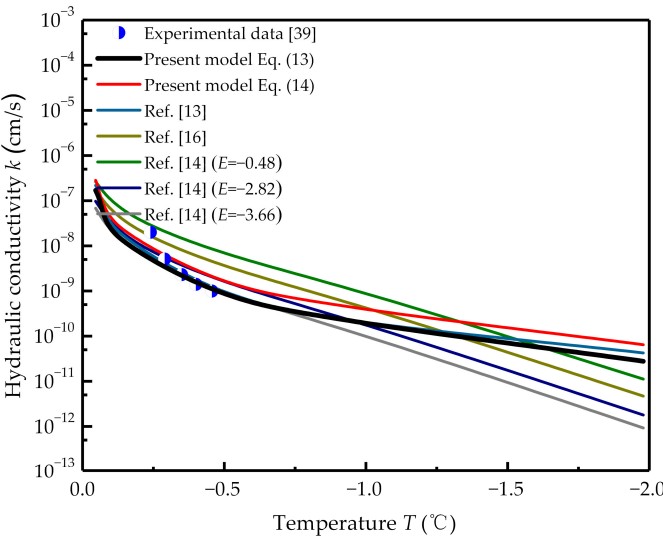

**Figure 10.** Comparison of the experimental data with different models for Tokoro [39] silt.

Table 3 summarises the MSE for the various models used to evaluate the model performance. As shown, the MSE value for the Nixon model is lowest, at 0.33, followed by the present model with Equation (13) where the MSE was 0.34. In contrast, the present model with Equation (13) is close to the Nixon model, which was determined by the experimental data. Using Equation (13) in the present model, the squared error at $-0.25\,°\text{C}$ was much higher than that at the other temperatures. This difference can be attributed to experimental error and model error. The latter error may be related to the cracks caused by frost heave. Cracks are preferential flow paths, which result in an increase in conductivities. This influence of frost heave on the pore structure was not considered in the fractal model; however, on the whole, the difference was small. The MSE value for the Mao et al. model

and Jame and Norum model ($E = -0.48$) was approximately 4 and 9 times larger, respectively, than for the present model with Equation (13). The MSE for the Jame and Norum model ($E = -2.82$ and $E = -3.66$) was close to that of the present model using Equation (14).

**Table 3.** Comparison results between various models.

| Model Type | Mean Squared Error | Squared Error | | | | |
|---|---|---|---|---|---|---|
| | | $-0.25\,°\text{C}$ | $-0.3\,°\text{C}$ | $-0.35\,°\text{C}$ | $-0.4\,°\text{C}$ | $-0.45\,°\text{C}$ |
| Present model Equation (13) | 0.34 | 1.62 | 0.08 | 0.001 | 0.009 | 0.01 |
| Present model Equation (14) | 0.41 | 0.55 | 0.08 | 0.33 | 0.54 | 0.55 |
| Nixon model [13] | 0.33 | 1.45 | 0.05 | 0.01 | 0.06 | 0.06 |
| Mao et al. model [16] | 1.37 | 0.03 | 0.80 | 1.55 | 2.14 | 2.32 |
| Jame and Norum model [14] $E = -0.48$ | 2.96 | 0.14 | 2.19 | 3.47 | 4.36 | 4.66 |
| Jame and Norum model [14] $E = -2.82$ | 0.40 | 0.84 | 0.02 | 0.22 | 0.46 | 0.48 |
| Jame and Norum model [14] $E = -3.66$ | 0.39 | 1.75 | 0.12 | 0.0007 | 0.03 | 0.04 |

In contrast, the fractal model provided a good agreement with the hydraulic conductivity data overall. Moreover, comparison with other models revealed the following advantages of the fractal model: fewer parameters; parameters are easy to obtain; each of the parameters has a clear physical meaning; and no measured conductivity is required in the model. Additionally, the fractal model can explain the reason for hydraulic conductivity changing with temperature; however, the influence of frost heave on the pore structure of frozen soil is not considered in the model, which may result in underestimating of the experimental data.

The present model has some limitations. Firstly, the model may be not valid in sandy soils, as these contain massive large pores that will freeze once temperature is lower than zero degrees Celsius, according to the SFC and Gibbs-Thomson equation. Second, the model does not consider the influence of the difference in SFC determined by the different methods on the hydraulic conductivity of saturated frozen soil. We expect this effect would not be pronounced when estimating the hydraulic conductivity, as hydraulic conductivity largely depends on pore radius. Third, the model does not consider the influence of frost heave on the pore structure of frozen soil, which may result in underestimating the experimental data. We expect this effect to be less pronounced in high-temperature frozen soil, where only slight frost heave occurs. Fourth, the model does not consider the influence of closed pores on hydraulic conductivity, which will result in overrating the hydraulic conductivity. Lastly, the model is only valid for soils fulfilling the fractal laws.

*6.2. Comparison of the Cumulative Pore Size Distribution Determined by the SFC and Mercury Intrusion Porosimetry (MIP)*

The SFC was used to determine pore size distribution in the present study. To verify this method, MIP was compared with this method. You et al. [40] and Xiao [41] measured the pore size distribution of silty clay using MIP. Figure 11 represents the SFC of the two soil samples. The solid line represents the freezing curve of You et al. [40], and the dashed line represent that of Xiao [41].

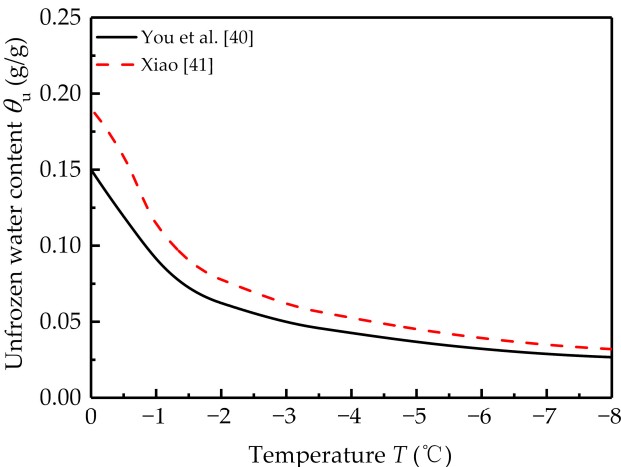

**Figure 11.** Soil freezing curves of the soils.

Figure 12 shows the comparison of pore size distribution determined by the SFC and MIP. The solid lines represent the pore size distribution determined by MIP and the dashed lines represent the pore size distributions determined by the soil freezing curve. It can be seen that the pore size distribution determined by the SFC captured the distinctive feature of the pore size distribution determined by MIP well; however, this method overrated the smaller pores measured by MIP. This result could be ascribed to the fact that the isolated pores—those that had no communication with the exterior of the sample could not be measured by MIP in any event, regardless of the pressure used [42], However, the isolated pores could be filled with water, which is considered in the pore size distribution determined by the SFC. Fagerlund [37] used a similar method to accurately predict the pore size distribution in the range 0.02 μm~0.5 μm for a certain sand-lime brick as shown in Figure 12c.

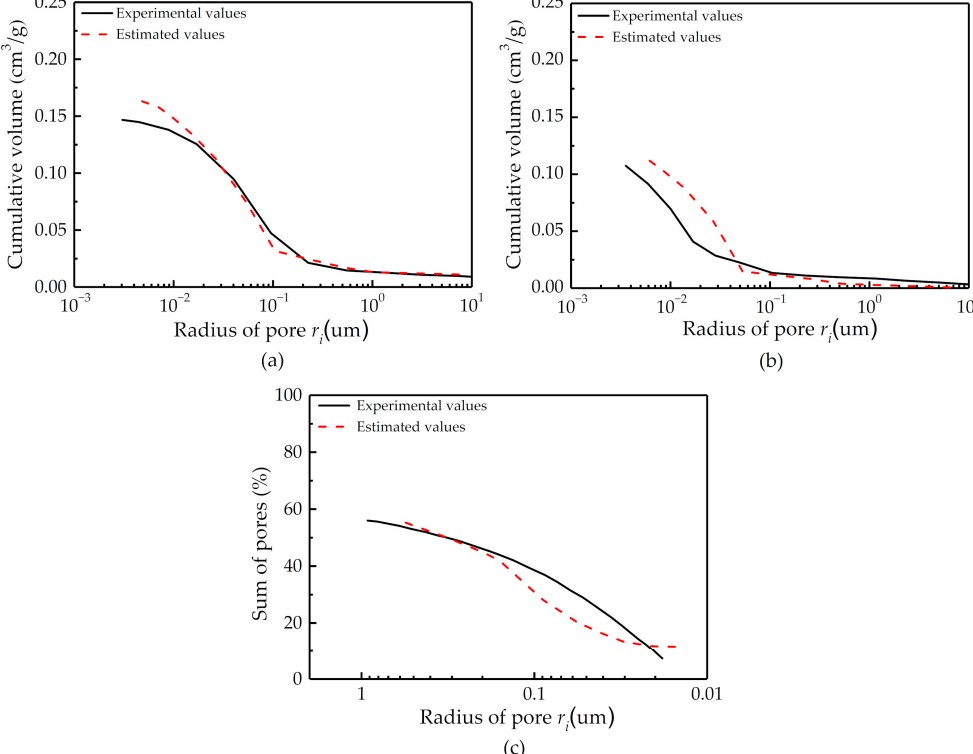

**Figure 12.** Comparison of the pore size distribution determined by SFC and by MIP. (**a**) Xiao [41]; (**b**) You et al. [40]; (**c**) Fagerlund [42].

### 6.3. Vvalidity of Darcy's Law

The present model was determined by Darcy's law. Although Darcy's law is widely applied to frozen soil, the validity of Darcy's law in frozen soil remains controversial. Several previous studies have shown that Darcy's law is applicable in frozen soil, at least in high-temperature frozen soil; for example, Burt and William [7] found a linear relationship between the hydraulic gradient and discharge for clayey silt within $-0.5\,^{\circ}$C using a laboratory experiment. Horiguchi and Miller [8] found the flux linearly changed with hydraulic conductivity for silt at different temperature within $-0.15\,^{\circ}$C. Tokoro [39] observed the same experimental phenomenon for silt within $-0.5\,^{\circ}$C.

To further account for the validity of Darcy's law, a theoretical proof based on the capillary bundle model and Reynolds number was proposed. The Reynolds number $R_e$ is always used to justify the flow state of the fluid in the capillaries

$$R_e = \frac{2r_{cr}\rho v}{\mu} \tag{21}$$

where $r_{cr}$ is the critical pore radius (m); $\rho$ is the density of the fluid (kg/m$^3$); and $v$ is the flow velocity (m/s). The flow velocity can be obtained by dividing water flow rate by the cross area of a single capillary

$$v = q/\pi r^2 = \frac{r^2}{8}\frac{\rho_w g}{\eta}\frac{\Delta H}{L_e} \tag{22}$$

Substituting Equation (22) into Equation (21), the critical pore radius $r_{max}$ for laminar flow was obtained as

$$r_{max} = \sqrt[3]{\frac{4R_e\mu^2\tau}{\rho_w^2 gi}} \tag{23}$$

The unfrozen water film is much smaller than the radius of capillaries; therefore, the geometry of ice in frozen soil is assumed to be identical to that of air in drying soil. Thus, the relationship between the matric potential of frozen soil and temperature can be estimated using the generalised form of the Clausius-Clapeyron equation, treating ice pressure the same as gauge pressure

$$h = \frac{L_f}{g}\ln\frac{T_m - \Delta T}{T_m} \tag{24}$$

where $|h|$ is the hydraulic head (m) and $\Delta T$ is the freezing temperature depression ($^{\circ}$C). Figure 13 presents the $|h|-\Delta T$ relationship obtained using Equation (24).

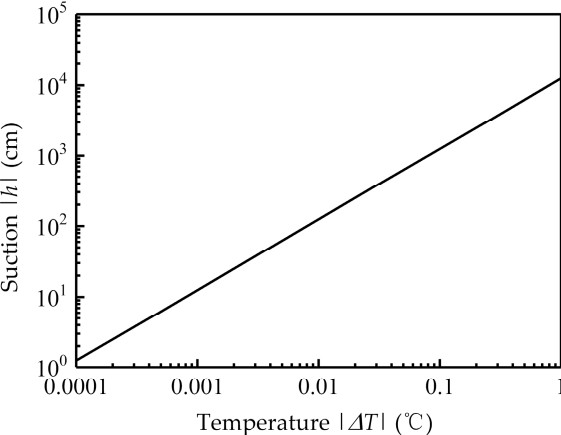

**Figure 13.** Relationship between the hydraulic head and temperature for frozen soil estimated using the Clausius-Clapeyron equation.

The hydraulic gradient $i$ is

$$i = \frac{|h|}{L} \tag{25}$$

where, $L$ is the length of soil column (m). Inserting Equation (19) into Equation (20) gives the expression for the hydraulic gradient

$$i = \frac{L_f}{Lg} \ln \frac{T_m - \Delta T}{T_m} \tag{26}$$

Substituting Equation (26) into Equation (23) gives the expression for the critical pore radius

$$r_{max} = \sqrt[3]{\frac{4 R_e \mu^2 \tau L}{\rho_w{}^2} \frac{1}{L_f \ln \frac{T_m - \Delta T}{T_m}}} \tag{27}$$

With $R_e = 2300$, $g = 9.81$ m/s$^2$, $\mu = 1.792 \times 10^{-3}$ Pa$\cdot$s, $\rho_w = 998$ kg/m$^3$, $L = 334.56$ KJ/kg, $T_m = 273.15$ K, $\Delta T = -1\,°C$ $L = 3$ cm and, $\tau = 2$, the critical pore radius is obtained:

$$r_{cr} = 2.95 \times 10^{-4} \sqrt[3]{L_e} \tag{28}$$

Taking $L = 3$ cm and $\tau = 2$ for example, the critical radius obtained is $r_{cr} = 116$ μm. That is, for a soil column with a length of 3 cm and tortuosity of 2, the water flow will be turbulent in pores larger than 116 μm at temperature differences of $-1\,°C$.

In the further test, the critical pore radius was compared with the pore size of soils determined by MIP from You [40], Xiao [41], Tao [43], Penumadu [44], and Juang [45]. As seen in Figure 14, the silt, silty clay, clay and Kaolin clay soils all had barely any pores with a radius larger than 116 μm. For sandy soils except for sand C (70% sand + 30% clay), they all had several pores with a radius larger than 116 μm. Thus, the validity of Darcy's law is reduced in sandy soils with larger pores. The critical radius increased with the increased length of soil sample and tortuosity, but decreased with the temperature difference increase. This method only presents the validity of Darcy's law for saturated frozen soil; for unsaturated frozen soil, the application of Darcy's law is reduced owing to the existence of the matrix potential.

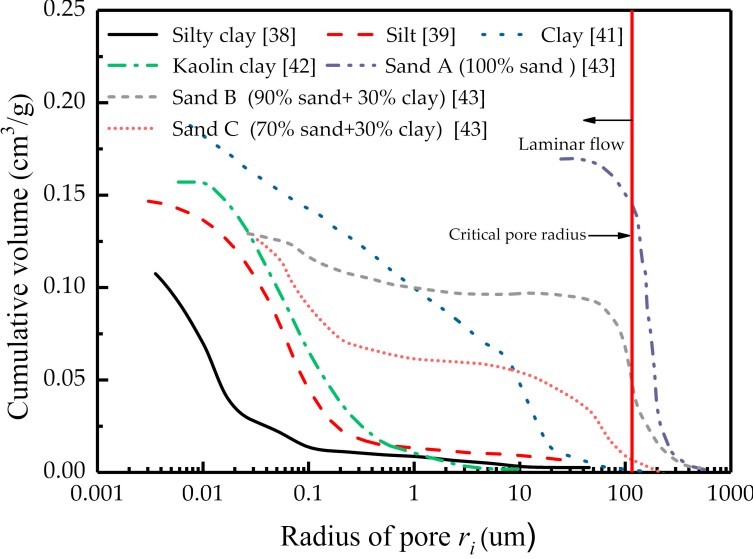

**Figure 14.** Pore size distribution of the soils.

## 7. Conclusions

In the present study, based on fractal theory and Hagen-Poiseuille's law, a fractal model of hydraulic conductivity for saturated frozen soil was developed. The model was verified using experimental results. The following conclusions can be drawn:

1. The model provides a good agreement with the hydraulic conductivity data. Compared with other models, this model is simpler to use owing to its simple requirements of the SFC, initial volumetric content of water and the dimensions of the soil column.
2. Pore size played a key role in affecting the hydraulic conductivity of frozen soil. The pore size dimension decreased linearly with temperature, the maximum pore size decreased with temperature and the tortuosity increased with temperature. Changes in these parameters with temperature can explain why the hydraulic conductivity of frozen soil changes with temperature.
3. The pore size distribution is approximately estimated by the SFC, making this method a possible alternative to MIP.
4. Darcy's law is valid in the saturated frozen silt, clayed silt and clay, but maybe not valid in saturated frozen sand and unsaturated frozen soil.

**Author Contributions:** Conceptualization, L.C. and F.M.; Formal analysis, L.C.; Methodology, L.C. and F.M.; Writing–original draft, L.C.; Writing–review & editing, D.L., F.M., X.S. and X.C.

**Funding:** This work was funded by the National Nature Science Foundation of China (No. 41701060 and No. 41271080), the funding of Key Research Program of Frontier Science of Chinese Academy of Sciences (QYZDY-SSW-DQC015), and the funding of the State Key Laboratory of Frozen Soil Engineering (No. SKLFSEZT17).

**Acknowledgments:** The manuscript was further improved by the insightful comments of anonymous reviewers from Water, which is greatly appreciate.

**Conflicts of Interest:** The authors declare no conflict of interest.

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
