# Peer review of "A Fractal Model of Hydraulic Conductivity for Saturated Frozen Soil"

_water, doi:10.3390/w11020369_

Round 1

Reviewer 1 Report

The article presents an attempt to find a method for determining the hydraulic conductivity of frozen soils on the basis of the fractal approach applicated to the pore size distribution. The paper could be an interesting contribution to our knowledge about soil freezing. However, in my opinion, it requires major revisions (honestly speaking I think that it should be reject but giving the authors another chance seems a good decision in this case). The most important disadvantages of this article can be listed as follows:

1. Juggling with parameters. The authors use equations, the main parameter of which (the fractal dimension D) is taken arbitrary, without any explanation. Furthermore, calculations are made with a value of this parameter which is not given to the reader. However, the knowledge about the value used in computations seems not very significant because anyway we do not know criteria to determine it for a real soil-water system. Of course determination of the pore size distribution curve can help with this, but this raises the obvious question why not determine the hydraulic conductivity directly. As seen, such a model is useless from practical point of view, in contrast to the authors’ assertions.

2. The validation of the model is strongly based on the results of other researchers. Interestingly, any own investigations have not been carried out. Instead, experimental data from different publications are used for comparisons. There would be nothing wrong with that, however, what about the method of selecting the sources? This brings to mind, by analogy to juggling with parameters, juggling with experimental data.

3. There are errors in the model derivation.

4. Any assumptions or limitations are not given. For example, the authors applicate the Darcy’s law not saying anything about its limitations. In particular, it is not clear if this law, originally formulated for sands at temperatures above 0oC, is still applicable to the flow of water in partially frozen capillaries. It is known that the latter involves factors other than just the pressure gradient, particularly at the heaving process frequently mentioned by the authors. Similarly, it is not known for which types of soils the model is applicable (it is clear that not to all).

More detailed comments are as follows:

37-38 “The unfrozen water transports in frozen soil follows the Darcy’s law”.

 The term “the unfrozen water” is ambiguous. A distinction between the unfrozen water coexisting with ice in frozen soil and the water which has not frozen yet, the latter existing in the zone below the freezing zone, should be done. Actually, the flow of the former is strictly limited and surely is not governed by the Darcy’s law, in contrast to the flow of the water from below to the zone of freezing.

84-89 This fragment should be written once again to make it more clear. Particularly, who are “they” mentioned in the text? What is a short story of the fractal approach?

126 “Since 1 < D < 2 , the exponent 4 -D < 2”

It seems that just the contrary, since 1 < D < 2 , the exponent 4 -D > 2

126 “ ”

A sign has been lost before 1. However, if the root is less than 10-2, the value of this expression can be close to 1 when 4 – D < 2 (for example 0.00990.001 = 0.99 and the value on the right side of Eq. (13) becomes close to 0. Therefore the statement 4 – D < 2 must be carefully examined.

Eqs. (18) and (19) How the porosity ε here is defined? If just as the ratio of pores to the total volume of soil, these equations make limited sense. There is a very distant relationship between the porosity and hydraulic conductivity. Strictly speaking, the hydraulic conductivity depends on pore throat radii, not on porosity itself. For example, clays typically have very low hydraulic conductivity (due to their small pore throat radii) but also have very high porosities (due to the structured nature of clay minerals). Of course the D-value may be affected by some special features of a given system. Unfortunately, there is no mention about this in the paper.

Eq. (18) What about the tortuosity τ absent in this equation but mentioned below it?

Eq. (19) Once again, the Darcy’s law has its strict limitations and assumptions (the Reynolds number!). The authors of the paper under review do not give them. Does it mean that the proposed solution is valid for all possible soil-water systems?

138-142 The conclusions here are partially false in the light of the common knowledge on soil-water systems. In particular, the hydraulic conductivity does not depend on porosity, which was shown above taking clays as an example.

187-188 “Figure 3 presents the soil freezing curves of the three soil samples. As shown, the freezing curve decreases dramatically near 0 , after which the freezing curve becomes gradual.”

These are banal observations. What would be interesting is to find out which method was used to determine the unfrozen water content curves. The problem is not trivial and it is widely known that adjusting different methods to the same soil material gives different results whereas the authors seem to regard these data as absolutely correct.

173-176: “In this model, the porosity is the ratio between the total volume of capillaries and the volume of the soil column. Because the unfrozen capillaries are assumed to be fully filled with unfrozen water, so the volume of unfrozen water can be conceived as the volume of the capillaries. Thus, the porosity is equivalent to the volumetric content of unfrozen water”.

This fragment raises serious doubts. What about the ice in some of capillaries during the freezing process? What this statement has to do with the scheme illustrated in Fig. 2?

203-204 “Figure 5 (a) presents the relationship between the pore size dimension and temperature, which is computed by the cumulative pore size distribution using the Eq. (23).”

Using Eq.(23) requires exact value of D. However, only an approximated range has been given in the paper (1 < D < 2). How the D-value was determined here? What is the rule for probable users of the model? Of course the authors could  adjust D to fit the experimental data best, but this is nothing but juggling with parameters.

Figure 6 and the related discussion. The obtained results are of course true but only qualitatively. To predict these facts, no model is needed, this is common knowledge. However, the authors promised to present a model able to predict exact and realistic values of hydraulic conductivity, which was to be easier than making direct experiments. The data in Fig. (6) are exact indeed, yet there are serious doubts if they are true (please mind the difference).

Figure 7 and the related discussion. Both Eq. (19) and Eq. (20) requires exact value of D, which cannot be predicted without quite a complicated laboratory investigation (and some of them may not fulfill the fractal rule). As stated above, juggling with values of this parameter can approximately fit the theoretical and experimental curves, but what a user of the model can do with it?

Author Response

Dear professor,

   First, thank you very much for giving us an opportunity to revise our manuscript. Second, thanks very much for your critical comments and suggestions on our manuscript titled “ A fractal model of hydraulic conductivity for saturated frozen soil”. We have addressed the comments and suggestions to the best of our abilities. We sincerely hope you will be satisfied with this responses. We highlighted the revised portions in red using the “Tracking Changes” function in the manuscript. The point-by-point responses are given in the attachment.

Thank you again for your valuable time and critical comments and suggestions.

Best regards for your!

Yours sincerely

Chen Lei, Li Dongqing, Ming Feng, Shi Xiangyang, Chen Xin

State Key Laboratory of Frozen Soil Engineering, Northwest Institute of Eco-Environment and Resources, Chinese Academy of Sciences, Lanzhou 730000, China;

University of Chinese Academy of Sciences, Beijing 100049, China;

Reviewer 2 Report

Chen et al. detail what appears to be a novel approach for estimating the hydraulic conductivity of frozen soil using a fractal approach. The method seems to be useful and nicely integrates with the soil freezing curve. This is a major research topic now in cold regions hydrology since the Arctic and Subarctic are beginning to thaw and there is considerable interest in how that may alter landscape hydraulic conductivity. I believe there is something valuable in this manuscript, but it is not presented very well, so it was hard for me to review it in detail.

Major suggestions

This needs major English editing to be more understandable. I began this process and gave up.

The layout for this paper is extremely odd. As examples:

The introduction has 10 equations (including 8a and 8b). Most of these are unnecessary. The introduction should not be used to reproduce endless equations but rather to introduce the reader to why they should care about the hydraulic conductivity of frozen soil. This is not sufficiently established.

The Results section includes Figure 3, which does not contain results from this study

Also, Table 1 could be better located in a Methods section

It is not entirely clear how this advances beyond the capillary bundle model of Watanabe and Flury. I'm sure the authors understand this, but it should be made clearer to someone who is not an expert in the field. Is it the fractal distribution assumption or the relationship to the soil freezing curve for parameterization?

Related to point 3, the authors introduce 'fractal model' (L84) without even explaining what it is or why it is important.

Figure 2 - why couldn't a smaller pore begin to fill with ice before a larger pore was completely (except for residual water) frozen? In other words, under this model, if there was a pore that was 1.0 mm in diameter, and one that was 1.00001 mm in diameter, would the smaller one have to completely freeze before the next one would begin to freeze? The figure suggests this.

A flow chart showing how the soil freezing curve were used to generate the pore size distribution and the K etc. would be useful for this paper. It is incredibly confusing to try to understand how everything ties together.

Equation questions:

Why does Eq. 1 not consider elevation contributions to hydraulic energy? 

I dislike the presentation of equations without corresponding units, and this is especially true when the equations are empirical (see intro)

Eqs. 8-9 are not presented in such a way as to be understandable

L132, the calculation of obtaining the seepage velocity  from the average velocity does not require the 'Dupuit Forcheimer Equation'

L193-194, what equations were used for this?

Could the method be extended for unsaturated conditions based on assumptions about the pore air/water/ice geometries? 

L156-165, this could be helped with a diagram

 Figure 5, this is not explained very well. And the trends are actually not exponential as stated.

Minor issues

The use of seepage velocity throughout is unfortunate. That is more of a geotechnical engineering term, and simply isn't common in the cold regions hydrology/hydrogeology world that I think this paper is targeted at. I would call it Darcy flux or something like that.

Author Response

(The authors gave the same response as above.)

Reviewer 3 Report

I recommend to use a synonym for the word "agreement", which is very often used in the paper and it is a bit too repetitive.

Overall comments: The article presents a new mathematical model to estimate the hydraulic conductivity of frozen soil, respectively a fractal model. The parameters used by the fractal model are fewer and easy to obtain.   One important conclusion of the research was the pore size played a key role in affecting the hydraulic conductivity of frozen soil. Also, the fact that Darcy’s law is valid in the saturated frozen silt, clayed silt and clay, but maybe not valid non saturated frozen sand and unsaturated frozen soil. The model was tested in comparison with other three different models. The authors present the limitations of the model.

The paper is well structured, the aim of the research was presented. The discussions are in comparison with other researches and results. The equations presented are very thoroughly described, the figures are helpful in understanding the comparisons or relationships between parameters or models. The references are adequate to the subject and new. The conclusions are clear and concise.

Therefore, I recommend the article to be published in present revised form.

Round 2

Reviewer 2 Report

General comments:

The other reviewer understands capillary hydraulics better than I do and makes some valuable comments. I will let them address the physics/mathematics of the approaches. The presentation of the paper has improved as most of the material is now at least in an appropriate section and the logic of information flow (in the text and figures) is easier to follow. There are still some issues.

Specific comments

1.      From last time: The readability of this paper would still benefit from extensive English editing. I’m not sure what the journal’s policies are on that as I often see papers published in Water that frankly could use some editing

2.      From last time: I still don’t understand how equation 1 considers elevation head. The authors responded that elevation was considered in the pore water pressure. I’m not sure if we are referring to different things (maybe a discipline difference) but pressure head and elevation head are distinct. Pw = rho*g*h is a hydrostatic consideration and fundamentally is still referring to water depth not elevation. If h is elevation pressure would go up with elevation (which of course it does not). Anyway this is not the form of Darcy’s law that the average reader of Water would be familiar with.

3.      L43 – Darcy flux is not in units of volume/time

4.      The introduction is better but still not really well framed for emphasizing the importance of hydraulic conductivity (the focus is more on frozen soils). Connecting the discussion to hydrology would be key for example. See Walvoord and Kurylyk (2016, Vadose Zone Journal) and Kurylyk and Watanabe (2013, Advances in Water Resources) for recent high-level reviews. I likely reveal myself by these suggestions, but the authors should work at providing better context. These reviews need  not be cited, but the papers therein may be valuable. A huge missing component in this study is the need for new state-of-the-art cold regions hydrology/hydrogeology models that can be parameterized for frozen soil. If the authors’ approach works, it could be a game changer for these types of models.

5.      L119 – this method always overestimates the K – strong statement, please provide citations to support this

6.      L120, the fractal method is still not explained in the methods. The authors jump into listing different studies without fundamentally explaining what it is. At least they should include the text ‘See section 3’ to point the reader forward. On a related note, surely Section 3.1 warrants citations.

7.      From before: I understand that largest pores freeze first. That makes sense. My question was whether or not a large pore would completely freeze across the diameter prior to the initiation of freezing in a smaller pore. The figures suggest that’s the case (I may misinterpret something), but I highly doubt it is in nature. This is especially the case if freezing occurs radially as the authors suggest.

8.      From before: The manuscript would definitely benefit from incorporating the flow chart that was provided in response to me. Unless one is an expert on frozen soil hydraulics (arguably a global population of about 20 people), this paper will be hard to understand, especially with respect to how the different equations fit together.

Author Response

Dear professor,

   First, thank you very much for giving us an opportunity to revise our manuscript. Second, thanks very much for your critical comments and suggestions on our manuscript titled “A fractal model of hydraulic conductivity for saturated frozen soil”. We have addressed the comments and suggestions to the best of our abilities. We sincerely hope you will be satisfied with this response. We highlighted the revised portions in red using the “Tracking Changes” function in the manuscript. The point-by-point responses are given in attachment.

Thank you again for your valuable time and critical comments and suggestions.

Best regards!

Yours sincerely

Chen Lei, Li Dongqing, Ming Feng, Shi Xiangyang, Chen Xin

State Key Laboratory of Frozen Soil Engineering, Northwest Institute of Eco-Environment and Resources, Chinese Academy of Sciences, Lanzhou 730000, China;

University of Chinese Academy of Sciences, Beijing 100049, China;

Feb. 6, 2019
